# A NEURO-SYMBOLIC FRAMEWORK FOR ANSWERING CONJUNCTIVE QUERIES

## ABSTRACT

The problem of answering logical queries over incomplete knowledge graphs is receiving significant attention in the machine learning community. Neuro-symbolic models are a promising recent approach, showing good performance and allowing for good interpretability properties. These models rely on trained architectures to execute atomic queries, combining them with modules that simulate the symbolic operators in queries. Unfortunately, most neuro-symbolic query processors are limited to the so-called *tree-like* logical queries that admit a bottom-up execution, where the leaves are constant values or *anchors*, and the root is the target variable. Tree-like queries, while expressive, fail short to express properties in knowledge graphs that are important in practice, such as the existence of multiple edges between entities or the presence of triangles.

We propose a framework for answering arbitrary conjunctive queries over incomplete knowledge graphs. The main idea of our method is to approximate a cyclic query by an infinite family of tree-like queries, and then leverage existing models for the latter. Our approximations achieve strong guarantees: they are *complete*, i.e. there are no false negatives, and *optimal*, i.e. they provide the best possible approximation using tree-like queries. Our method requires the approximations to be tree-like queries where the leaves are anchors or existentially quantified variables. Hence, we also show how some of the existing neuro-symbolic models can handle these queries, which is of independent interest. Experiments show that our approximation strategy achieves competitive results, and that including queries with existentially quantified variables tends to improve the general performance of these models, both on tree-like queries and on our approximation strategy.

## 1 INTRODUCTION

Knowledge graphs play a crucial role in representing knowledge within organizations and communities. Their usage is now widespread both in industry and in the scientific community (Fensel et al., 2020; Hogan et al., 2021). Knowledge graphs model information as nodes, which represent entities of interest, and edges, that represent relations between entities. During the creation of knowledge graphs, however, information may be stale or conflicting, and certain sources of data may have not been integrated yet. As a consequence, knowledge graphs tend to be *incomplete* in the sense that some of the entities or relations occurring in the application domain may not be present in the graph. We refer to Ren et al. (2023) for statistics about missing information in knowledge graphs.

A particularly important reasoning task on knowledge graphs is the *answering of queries*. Traditional query answering methods, especially those from the data management and semantic web literature, focus on only extracting the information that can be derived from the knowledge *present* in the graph (Angles et al., 2017; Hogan et al., 2021; Ali et al., 2022). Given the incomplete character of knowledge graphs, these methods hence fail to address the need to reason about unknown information. This limits their usefulness in many application domains (Nickel et al., 2015).

This observation has spurred the development of numerous machine learning approaches to query answering, e.g. Hamilton et al. (2018); Ren et al. (2019); Ren & Leskovec (2020); Zhang et al. (2021); Zhu et al. (2022). We focus on a recently proposed family of approaches, namely, the *neuro-symbolic* models. They rely on trained (e.g. neural) architectures to execute atomic queries, and combine them with modules that simulate the symbolic logical operators in queries. These

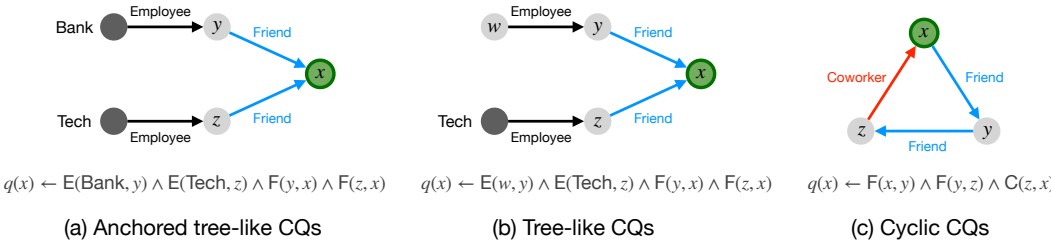

$$q(x) \leftarrow \mathsf{E}(\mathsf{Bank}, y) \wedge \mathsf{E}(\mathsf{Tech}, z) \wedge \mathsf{F}(y, x) \wedge \mathsf{F}(z, x)$$

(a) Anchored tree-like CQs

$$q(x) \leftarrow \mathsf{E}(w, y) \wedge \mathsf{E}(\mathsf{Tech}, z) \wedge \mathsf{F}(y, x) \wedge \mathsf{F}(z, x)$$

(b) Tree-like CQs

$$q(x) \leftarrow \mathsf{F}(x, y) \wedge \mathsf{F}(y, z) \wedge \mathsf{C}(z, x)$$

(c) Cyclic CQs

Figure 1: Different conjunctive queries (CQs) (a) atoms in anchored tree-like CQs are structured as trees where the leaves are anchors and the root the target variable ($x$ in this case); (b) leaves in tree-like CQs can be anchors or existential varibles ($w$ in this case); (c) arbitrary CQs can have cycles.

approaches have shown promising performance and, more importantly, produce more interpretable models. We refer to Ren et al. (2023) for a comprehensive recent survey on neural and neuro-symbolic approaches for query answering over incomplete knowledge graphs.

State-of-the-art neuro-symbolic approaches, however, only support a restricted class of queries, namely, *anchored tree-like* queries[1] (Ren et al., 2023). Figure 1a shows an example of an anchored tree-like query. Although tree-like queries already capture interesting properties in graphs, they are not capable of checking more complex properties such as the existence of triangles or of multiple edges between entities. The development of neuro-symbolic approaches for more complex query classes remains largely unexplored. In particular, supporting cyclic queries, such as the triangle query, has been identified as an important open challenge by Ren et al. (2023). Figure 1c shows an example of a cyclic (triangle) query. **In this paper we propose a neuro-symbolic framework for approximating complex queries by maximally leveraging methods for tree-like queries.**

More specifically, our **contributions** are as follows. **(1)** We propose an *approximation scheme* for complex conjunctive queries using tree-like queries. Moreover, the approximation scheme comes with *theoretical guarantees*: It is *complete* in the sense that no false negative query answers are produced. It is *optimal* in that we provide the best possible approximation using tree-like queries. **(2)** The approximation scheme is *adaptive* in the sense that it is parameterized by the notion of *depth* of tree-like queries. For any depth, an approximation exists and higher depth queries potentially provide better approximations. The choice of depth can be tuned depending on available resources, queries and data at hand. **(3)** Our approach is *generic* and can be used in combination with any neuro-symbolic query processor, provided that unanchored *tree-like queries* are supported. Figure 1b depicts an unanchored tree-like query in which the input node $w$ is variable. As an independent contribution, we show how to go from anchored to (unanchored) tree-like queries in some neuro-symbolic methods. **(4)** We implemented our approach on top of the GNN-QE implementation by Zhu et al. (2022). Results show our techniques are a viable strategy for answering cyclic queries, and that our improvements can be carried over with little cost over this standard neuro-symbolic architecture.

## 2 RELATED WORK

**Neural and neuro-symbolic query answering.** The machine learning community has produced a wide body of literature investigating how to answer complex queries over incomplete knowledge graphs. These works build on and extend recently successful methods desgined for knowledge graph completion (Bordes et al., 2013; Yang et al., 2015; Trouillon et al., 2016; Sun et al., 2019; Schlichtkrull et al., 2018; Vashishth et al., 2020; Teru et al., 2020). Following Ren et al. (2023), we can identify two different approaches to complex query answering. Firstly, *neural* approaches (Hamilton et al., 2018; Kotnis et al., 2021; Liu et al., 2022; Pflueger et al., 2022) answer queries by processing atomic queries and logical operators directly in the embedding space, parameterizing them with neural networks. These methods usually lead to better performance, but at the cost of being much less interpretable. Secondly, there are so-called *neuro-symbolic* approaches, which

---

[1]These queries are also referred simply as *tree-like* in the literature. We reserve the term tree-like for the generalization where the anchored condition is lifted.

combine neural approaches to compute missing links between entities and symbolic approaches to extract answers from the completed data (Bai et al., 2023; Luo et al., 2023; Chen et al., 2022; Ren & Leskovec, 2020; Yin et al., 2023; Zhu et al., 2022). While logical operators are still processed in the latent space, they are biased to better correlate with their symbolic counterparts. We refer to Ren et al. (2023) for more details on the particular workings of each of these models. To our knowledge, none of these approaches deal with general CQs.

**Approximation of conjunctive queries.**    The notion of a tree-like approximation of a conjunctive query, as explored in this paper, was originally introduced by the database theory community. Two types of approximations were proposed: *underapproximations*, which yield sound but not necessarily complete answers (Barceló et al., 2014), and *overapproximations*, which yield complete but not necessarily sound answers (Barceló et al., 2020). For reasons explained above, in this work we focus on overapproximations, that is, complete approximations of conjunctive queries. The main distinction between our work and previous research is the fact that tree-like approximations are evaluated using a neuro-symbolic approach. Additionally, we present the first working implementation of the concept of CQ approximation, as prior work had only examined its theoretical properties. Finally, previous works deal with a slighly different notion of tree-like, namely, *treewidth-1* queries, and hence some refinements are needed to obtain our theoretical results.

## 3 Preliminaries

**Knowledge graphs and conjunctive queries.**    Knowledge graphs are directed graphs with labeled edges. Formally, let $\mathsf{Con}$ be a countably infinite set of *constants*. A *knowledge graph* (KG) is a tuple $\mathcal{G} = (\mathcal{E}, \mathcal{R}, \mathcal{S})$ where $\mathcal{E} \subseteq \mathsf{Con}$ is a finite set of *entities*, $\mathcal{R}$ is a finite set of *edge types*, and $\mathcal{S} \subseteq \mathcal{E} \times \mathcal{R} \times \mathcal{E}$ is a finite set of *edges*. We typically denote an edge $(a, R, b)$ by $R(a, b)$.

Let $\mathsf{Var}$ be a countably infinite set of *variables*. As is common in machine learning, we focus on unary queries, that is, queries with only one target variable. Formally, a *(unary) conjunctive query* (CQ) $q$ over a set of edge types $\mathcal{R}$ is a first-order logic (FO) formula of the form

$$q(x) \leftarrow R_1(y_1, z_1) \wedge \cdots \wedge R_m(y_m, z_m),$$

where $x$ is the *target* variable, each $R_i(y_i, z_i)$ is an *atom* with $R_i \in \mathcal{R}$ and $\{y_i, z_i\} \subseteq \mathsf{Con} \cup \mathsf{Var}$ ($y_i, z_i$ are either variables or constants). The variable set $\mathsf{Var}(q)$ of $q$ is the set of variables appearing in the atoms of $q$, that is, the variables appearing in $\{y_1, z_1, \ldots, y_m, z_m\}$. Similarly, we denote by $\mathsf{Con}(q)$ the constants appearing in the atoms of $q$. As usual, we assume $x \in \mathsf{Var}(q)$. The variables in $\mathsf{Var}(q) \setminus \{x\}$ are the *existentially quantified* variables of $q$. Sometimes we write $q(x)$ instead of $q$ to emphasize that $x$ is the target variable of $q$. The semantics of CQs is defined using the standard semantics of first-order logic. We denote by $q(\mathcal{G})$ the *answer* of the CQ $q$ over the KG $\mathcal{G}$.

Figure 1c shows the CQ $q(x) \leftarrow \mathrm{Friend}(x, y) \wedge \mathrm{Friend}(y, z) \wedge \mathrm{Coworker}(z, x)$ looking for all persons $x$ that have a friend $y$ and a coworker $z$ that are friends with each other. Here, $\mathsf{Var}(q) = \{x, y, z\}$, $x$ is the target variable, and $y$ and $z$ are both existentially quantified.

The *query graph* of a CQ $q$ is the multigraph that has $\mathsf{Var}(q) \cup \mathsf{ConOcc}(q)$ as nodes, and an edge from node $u$ to node $v$ for every atom $R(u, v)$ in $q$. Here $\mathsf{ConOcc}(q)$ is the set of occurrences of constants in $q$, i.e. if the number of occurrences in different atoms of $q$ of a constant $a \in \mathsf{Con}(q)$ is $k$, then there are $k$ duplicates of $a$ in $\mathsf{ConOcc}(q)$. We say that a CQ $q(x)$ with target variable $x$ is *tree-like* if the query graph of $q$ is an (undirected) tree rooted in node $x$. In particular, no multiple edges between pair of nodes are allowed. Additionally, $q$ is *anchored* if all the leaves of this tree are nodes in $\mathsf{ConOcc}(q)$; otherwise it is *unanchored*. As we are working with $\mathsf{ConOcc}(q)$ instead of $\mathsf{Con}(q)$, different leaves could correspond to the same anchor. The *depth* of a tree-like CQ is the depth of the corresponding tree formed in its query graph, that is, the length of the longest path from the root to one of its leaves. Finally, $q$ is *cyclic* if the query graph of $q$ has an undirected cycle.

Figure 1 contains examples of anchored tree-like, tree-like and cyclic conjunctive queries, depicted using their query graph. Notice that the unanchored query in Figure 1b was obtained by existentially quantifying one of the leaves of the query in Figure 1a.

As we mentioned, most neuro-symbolic methods for logical query answering are restricted to *anchored tree-like* queries. Notice that one could define tree-like queries for the full FO fragment. This

is the fragment commonly dealt with in the literature, and our implementation also supports it. We refer to Ren et al. (2023); Yin et al. (2023) for the definitions.

**CQ containment.** A concept we will exploit heavily is that of query containment. We say that a CQ $q$ is *contained* in a CQ $q'$, denoted by $q \subseteq q'$, if $q(\mathcal{G}) \subseteq q'(\mathcal{G})$, for all KGs $\mathcal{G}$. That is, the answer of $q$ is always contained in the answer of $q'$, independently of the underlying KG. While this notion reasons over all KGs, it admits a simple syntactic characterization based on homomorphisms.

A *homomorphism* from CQ $q(x)$ to CQ $q'(x)$ is a mapping $h : \mathsf{Var}(q) \cup \mathsf{Con}(q) \to \mathsf{Var}(q') \cup \mathsf{Con}(q')$ from the variables and constants of $q$ to the variables and constants of $q'$ such that $h(x) = x$, $h(a) = a$ for all $a \in \mathsf{Con}(q)$, and $R(h(y), h(z))$ is an atom of $q'$, for all atoms $R(y, z)$ of $q$. That is, a homomorphism is a way of replacing the variables of $q$ by variables of $q'$ such that each atom of $q$ becomes an atom of $q'$. The target variable of $q$ must be mapped to the target variable of $q'$. The following is a well-known characterization of CQ containment.

**Proposition 3.1** (Chandra & Merlin (1977)). *A CQ $q$ is contained in a CQ $q'$ if and only if there is a homomorphism from $q'$ to $q$.*

## 4 ANSWERING CQS VIA TREE-LIKE APPROXIMATIONS

We now present our framework for answering arbitrary CQs over incomplete KGs. The idea of our method is to approximate a cyclic CQ $q$ by an infinite family $\mathcal{U}_q = \{\tilde{q}_d\}_{d \geq 1}$ of tree-like CQs. As already mentioned, by doing so we can use state-of-the-art neuro-symbolic methods – only designed for tree-like queries – to deal with complex queries as well. The family $\mathcal{U}_q$ is parameterized by the query depth: each $\tilde{q}_d$ is of depth $d$. By taking greater depths, we obtain better or equal approximations in $\mathcal{U}_q$. Interestingly, the family $\mathcal{U}_q$ provides us with the following formal guarantees:

- *Completeness:* CQs in $\mathcal{U}_q$ are *complete*, that is, their answers always contain the answer of $q$. In other words, $q$ is contained in each $\tilde{q}_d$ and hence $\tilde{q}_d$ does not produce false negatives.
- *Optimality:* For each depth $d \geq 1$, the CQ $\tilde{q}_d \in \mathcal{U}_q$ is the best approximation (in a precise sense) among all the complete tree-like approximations of $q$ with depth at most $d$.

This suggests the following **neuro-symbolic method for answering complex queries**: Take any neuro-symbolic method capable of answering tree-like queries. Then, given a CQ $q$, we answer $q$ by feeding one of its tree-like approximations $\tilde{q}_D \in \mathcal{U}_q$ (the chosen depth $D$ is a hyperparameter of our model) to the chosen neuro-symbolic method. We thus leverage existing neuro-symbolic methods for anchored tree-like CQs, both for inference and learning. We remark that current methods work with anchored tree-like CQs, while our approach requires the use of (not necessarily anchored) tree-like CQs. We show in Section 4.3 how to remedy this but first formalize our approach.

### 4.1 COMPLETE TREE-LIKE APPROXIMATIONS

Let $q$ be an arbitrary CQ. A *complete tree-like approximation* of $q$ is a tree-like CQ $q'$ that contains $q$. That is, the answer $q(\mathcal{G})$ is always contained in the answer $q'(\mathcal{G})$, independently of the KG $\mathcal{G}$. We stress that the notion of completeness is particularly relevant for the setting of incomplete KGs. Indeed, we are given a CQ $q$ and an (incomplete) KG $\mathcal{G}$ and the goal is to obtain the answers $q(\mathcal{G}^*)$ for the unobservable complete KG $\mathcal{G}^*$. As containment considers all possible KGs, the answer $q'(\mathcal{G}^*)$ of a complete approximation $q'$ must hence contain the sought answer set $q(\mathcal{G}^*)$.

By Proposition 3.1, a tree-like CQ $q'$ is a complete approximation of $q$ if there is a homomorphism from $q'$ to $q$. Of course, there could be many approximations for the same query. As an example, consider the triangle CQ $q(x) \leftarrow \mathsf{Friend}(x, y) \wedge \mathsf{Friend}(y, z) \wedge \mathsf{Coworker}(z, x)$ depicted in Figure 2. On the right of Figure 2, we can see three possible complete tree-like approximations for $q$. Indeed, $q'_1$ and $q'_2$ can be mapped to $q$ via the homomorphism $\{x \mapsto x, y \mapsto y, z \mapsto z, x' \mapsto x\}$. For $q'_3$, we can use the homomorphism $\{x \mapsto x, y_1 \mapsto y, y_2 \mapsto y, z_1 \mapsto z, z_2 \mapsto z, x_1 \mapsto x, x_2 \mapsto x\}$. Actually, by taking longer paths, it is easy to get infinitely many approximations for $q$. Hence, the space of approximations may be infinite. This raises the question which approximation should we choose? We discuss this problem in the next section.

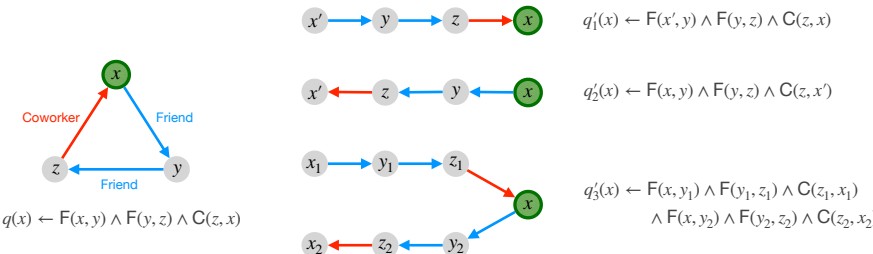

Figure 2: A cyclic CQ $q$ and three possible complete tree-like approximations. Best viewed in color.

## 4.2 COMPLETE OPTIMAL APPROXIMATIONS: UNRAVELINGS

While the number of approximations is infinite, we show there are a special kind of approximations that are optimal in a precise sense, and hence are a natural choice to approximate the original CQ.

Let $q(x)$ be a CQ. A *valid path* of $q(x)$ is a sequence $x_0, A_1, x_1, \ldots, A_k, x_k$, for $k \geq 0$, such that:

- $x_0 = x$, each $x_i \in \mathsf{Var}(q) \cup \mathsf{Con}(q)$, and each $A_i$ is an atom of $q$.
- for each $1 \leq i \leq k$, the atom $A_i$ is either of the form $R(x_{i-1}, x_i)$ (a forward traversal of the atom), or $R(x_i, x_{i-1})$ (a backward traversal of the atom).
- $A_i \neq A_{i+1}$, for each $1 \leq i < k$.

Intuitively, a valid path is a way of traversing the CQ $q$ starting from the target variable $x$ and sequentially moving through the atoms of $q$. We can visit the same variable, constant or atom several times. The only restriction is that an atom cannot be visited multiple times *consecutively* in the sequence. Hence, once an atom is traversed, we cannot go back via the same atom immediately. The *length* of a valid path is the number of atoms $k$. Note that the valid path of length $0$ is well-defined and corresponds to the sequence $x$. A valid path is *unanchored* if it ends at a variable of $q$; otherwise, we say that it is *anchored*. For a valid path $P$, we denote by $\mathsf{end}(P) \in \mathsf{Var}(q) \cup \mathsf{Con}(q)$ the element at the end of path $P$.

Consider the CQ $q(x) \leftarrow A_1 \wedge A_2 \wedge A_3$ in Figure 2, where $A_1 = \mathsf{Friend}(x, y)$, $A_2 = \mathsf{Friend}(y, z)$ and $A_3 = \mathsf{Coworker}(z, x)$. An example of an unanchored valid path is $x, A_1, y, A_2, z, A_3, x$, which corresponds to a clockwise traversal of length $3$ starting at $x$. The anticlockwise traversal of length $3$ is given by the valid path $x, A_3, z, A_2, y, A_1, x$.

Now we are ready to define our optimal approximations. Let $q(x)$ be a CQ. The *unraveling* of $q(x)$ of depth $d \geq 1$ is the tree-like CQ $\tilde{q}_d(x)$ constructed as follows:

- The variables of $\tilde{q}_d$ correspond to the unanchored valid paths of $q$ of length at most $d$. Formally, $\mathsf{Var}(\tilde{q}_d) := \{z_P \mid P \text{ unanchored valid path of } q \text{ of length} \leq d\}$.
- For valid paths $P$ and $P' = P, A', \mathsf{end}(P')$ of $q$ of lengths $\leq d$, if $A' = R(\mathsf{end}(P), \mathsf{end}(P'))$ then $\tilde{q}_d$ has an atom $R(o_P, o_{P'})$, where $o_W = z_W$ if $W$ is unanchored, and $o_W = \mathsf{end}(W)$ otherwise. If $A' = R(\mathsf{end}(P'), \mathsf{end}(P))$ then $\tilde{q}_d$ has an atom $R(o_{P'}, o_P)$.
- The target variable $x$ of $\tilde{q}_d$ is $z_{P_0}$, where $P_0$ is the valid path of $q$ of length $0$.

The idea is that the unraveling $\tilde{q}_d(x)$ of depth $d$ of $q(x)$ is obtained by traversing $q$ in a tree-like fashion, starting from the target variable $x$ and moving from one variable to all of its neighbors, through the atoms of $q$. Every time we add fresh variables to the unraveling and hence this is actually a tree-like CQ. The tree traversal has depth $d$ and is always restricted to valid paths (no immediate returns to the same atom). The leaves of the unraveling could be anchors or existentially quantified variables. The latter case is unavoidable in general and hence the need of working with (not necessarily anchored) tree-like CQs.

Continuing the example from Figure 2, $q_3'$ is the depth $3$ unraveling of $q$. Note how the variables $z_1$, $y_1$, $x_1$ of $q_3'$ correspond to the valid paths $(x, A_3, z)$, $(x, A_3, z, A_2, y)$, $(x, A_3, z, A_2, y, A_1, x)$. Similarly, the variables $y_2$, $z_2$, $x_2$ correspond to $(x, A_1, y)$, $(x, A_1, y, A_2, z)$, $(x, A_1, y, A_2, z, A_3, x)$.

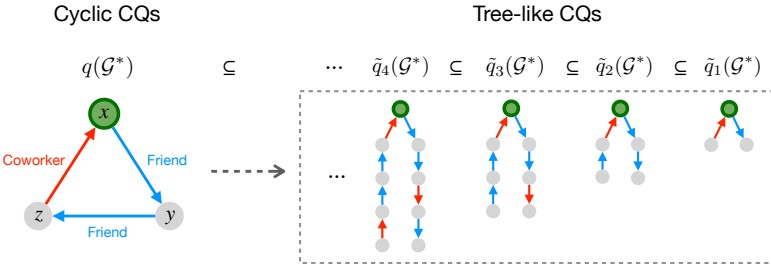

Figure 3: Overview of our approach. We show unravelings until depth 4. The goal is to approximate the answer $q(\mathcal{G}^*)$ of $q$ over the unobservable complete KG $\mathcal{G}^*$. Best viewed in color.

By definition, the unraveling $\tilde{q}_d$ is a tree-like CQ. By inverting the "unraveling" process, we can obtain a homomorphism from $\tilde{q}_d$ to $q$, and then $\tilde{q}_d$ is always a complete tree-like approximation. Also, we have that $\tilde{q}_{d+1} \subseteq \tilde{q}_d$ holds and hence the family $\mathcal{U}_q = \{\tilde{q}_d\}_{d \geq 1}$ provides potentially better approximations as depth increases ($q \subseteq \cdots \subseteq \tilde{q}_3 \subseteq \tilde{q}_2 \subseteq \tilde{q}_1$). These properties are summarized in the following proposition (see Appendix A for a formal proof).

**Proposition 4.1.** *Let $q$ be a CQ and $d \geq 1$. The unraveling $\tilde{q}_d$ is a complete tree-like approximation of $q$. Moreover, $\tilde{q}_{d+1} \subseteq \tilde{q}_d$ holds.*

Interestingly, we can show that $\tilde{q}_d$ is optimal in the following sense:

**Theorem 4.1.** *Let $q$ be a CQ and $d \geq 1$. Suppose $q'$ is a complete tree-like approximation of depth at most $d$. Then $\tilde{q}_d \subseteq q'$ holds.*

In particular, for any complete tree-like approximation $q'$ of $q$, there exists an unraveling $\tilde{q}_d$ at least as good than $q'$ as an approximation of $q$, i.e. $q \subseteq \tilde{q}_d \subseteq q'$. The proof idea of Theorem 4.1 is to turn any homomorphism $h$ from $q'$ to $q$ into a homomorphism from $q'$ to $\tilde{q}_d$ by analyzing the image of $h$ on $q$. See Appendix A for complete details. Figure 2 shows the depth 3 unraveling $\tilde{q}_3 = q'_3$ for $q$, and two additional depth 3 approximations $q'_1$ and $q'_2$. We see that $q \subseteq \tilde{q}_3 \subseteq q'_1$ and $q \subseteq \tilde{q}_3 \subseteq q'_2$.

In conclusion, we have shown that the tree-like queries $\mathcal{U}_q = \{\tilde{q}_d\}_{d \geq 1}$ satisfy the desired properties of completeness and optimality. Figure 3 shows an overview of our approach for the triangle query. We next show how to turn the theory into practice.

### 4.3 A CONCRETE IMPLEMENTATION: ∃GNN-QE

One of the key strengths of our proposed approximation scheme is that it is *generic*. That is, it can be implemented on top of any neuro-symbolic query processing method, provided that the method is capable of dealing with (possibly unanchored) tree-like queries.

This claim comes with a small caveat. As already mentioned, state-of-the-art methods deal with *anchored* tree-like queries (see also Ren et al. (2023)) and modifications are needed to support general tree-like queries. The challenge is to encode unanchored leaf nodes in tree-like queries in the latent space in which the encoding of anchored entities typically reside. Importantly, the encoding needs to simulate existential quantification in line with the semantics of unanchored leaf nodes. We here describe how this can be done in fuzzy-based neuro-symbolic approaches such as Chen et al. (2022); Zhu et al. (2022), leaving the extension of other approaches to future work.

Our implementation is based on GNN-QE by Zhu et al. (2022), a neuro-symbolic architecture that processes anchored tree-like queries in a bottom-up fashion. Anchored leaf nodes are encoded as one-hot vectors in latent space, and edges between entities are processed using an adaptation of the NBFNet graph neural network (Zhu et al., 2021). In each step, a probabilistic vector over entities is obtained, indicating the likelihood of an edge to those entities. Intuitively, the knowledge graph is completed by these edge predictions. Finally, the probability vectors are combined using operations that simulate logical operations. For example, for the query in Figure 1a, a one-hot vector encoding anchor "Tech" is transformed through the Employee edge into a vector indicating the probability that someone (entity) works in Tech. Following this reasoning, *we encode unanchored leaf nodes as full unitary vectors* (that is, a vector consisting of all ones). Such a vector indeed gives equal

probability to every entity hereby simulating existential quantification. For example, to answer a query such as the one in Figure 1b, we encode the $w$ node by the full unitary vector, the anchor node "Tech" remains encoded as before. We then process these vectors as in GNN-QE. We denote our extension by ∃GNN-QE. Since it can deal with general tree-like queries, we can use it alongside our approximation scheme. In the next section we report how well everything works in practice.

## 5 EXPERIMENTS

We set up our experimental evaluation to address the following questions. The first two questions relate to our support for general (not necessarily anchored) tree-like queries.

(**Q1**) What is the effect on the performance of answering anchored tree-like queries when the training set includes unanchored tree-like queries as well?

(**Q2**) Similarly, what is the effect on the performance of answering general tree-like queries?

Looking ahead, as a contribution of independent interest, our results indicate that we can support general tree like queries (**Q2**) with little or no negative impact for both anchored tree-like queries (**Q1**). This gives us ground to suggest that general tree-like queries should become default members in training and testing sets of future neuro-symbolic architectures.

Our third question relates to our approximation scheme.

(**Q3**) What is the performance of our approximation scheme in answering cyclic queries? And related, how does this depend on the chosen depth of the unravelling?

Looking ahead, our results show that unravellings can be used to answer cyclic queries: the metrics obtained for our suite of cyclic test queries are competitive, albeit slightly weaker, with similar metrics obtained by recent approaches for complex tree-like queries involving unions and negations of atoms. We thus validate the potential of our approach and promote it to become a competitive standard for future algorithms dealing with complex query types.

### 5.1 EXPERIMENTAL SETUP

We perform our evaluation on the commonly used *knowledge graphs* FB15k-237 (Toutanova & Chen, 2015), FB15k (Bordes et al., 2013) and NELL995 (Xiong et al., 2017) with their official training/validation/testing split. With regards to methods, as *baseline* we use GNN-QE, trained following the specifications of Zhu et al. (2022). That is, it is trained using the queries generated by BetaE (Ren & Leskovec, 2020), consisting of 10 tree-like query types, including queries featuring union and negation (1p/2p/3p/2i/3i/2in/3in/inp/pni/pin). For our method[2] ∃GNN-QE we additionally provide a new set of training, validation and test queries without anchors and unravelings of cyclic queries alongside with their corresponding answers for FB15k-237, FB15k and NELL995. These queries adhere to the same query types as before, except they are not anchored. In order to ensure a fair comparison, we trained ∃GNN-QE keeping the training parameters identical to those used for GNN-QE, but including queries without anchors. Details and statistics of both the new query set and the training can be found in the Appendix B. Metrics of GNN-QE are taken from its original paper by Zhu et al. (2022). Specifically, we report the *mean reciprocal rank* (**mrr**) of the predicted answer set (compared with the ground truth), and the *Spearman correlation rank* (**spearmanr**) between the total number of answers predicted and the number of answers in the ground truth. We report the remaining metrics used in Zhu et al. (2022) in Appendix B. Results are measured only against GNN-QE, see their original paper for comparison against other methods.

### 5.2 RESULTS

**Anchored tree-like queries.** In our first batch of experiments we investigate the *effect of training with unanchored queries* on the performance on the original *anchored* BetaE queries (**Q1**). We compare the performance of GNN-QE and ∃GNN-QE on anchored queries on our datasets. Importantly, as mentioned already, GNN-QE is trained using the original BetaE queries, whereas ∃GNN-QE is trained using additional unanchored BetaE queries. In Table 1 we report the results.

---

[2]Code is available at https://anonymous.4open.science/r/exists-gnn-qe-8A7C/

Table 1: Mean reciprocal rank and spearman rank correlation on test BetaE queries. Results of GNN-QE are taken from (Zhu et al. (2022)). Other metrics can be found in Appendix C

| Metric | Model | 1p | 2p | 3p | 2i | 3i | ip | pi | 2in | 3in | inp | pin | pni | 2u | up |
|---|---|---|---|---|---|---|---|---|---|---|---|---|---|---|---|
| | | | | | | | FB15k-237 | | | | | | | | |
| spearmanr | GNN-QE | 0.948 | 0.951 | 0.895 | 0.992 | 0.970 | **0.937** | 0.911 | 0.981 | 0.968 | **0.864** | 0.880 | 0.987 | - | - |
| | ∃GNN-QE | **0.977** | **0.966** | **0.942** | 0.992 | **0.975** | 0.898 | **0.943** | **0.990** | **0.981** | 0.853 | **0.933** | **0.989** | 0.979 | 0.968 |
| mrr | GNN-QE | **0.428** | **0.147** | **0.118** | **0.383** | **0.541** | **0.189** | **0.311** | **0.100** | **0.168** | **0.093** | **0.072** | **0.078** | **0.162** | **0.134** |
| | ∃GNN-QE | 0.321 | 0.107 | 0.096 | 0.339 | 0.501 | 0.147 | 0.268 | 0.063 | 0.139 | 0.080 | 0.053 | 0.048 | 0.119 | 0.103 |
| | | | | | | | FB15k | | | | | | | | |
| spearmanr | GNN-QE | 0.958 | **0.970** | **0.940** | **0.984** | 0.927 | **0.916** | **0.936** | 0.980 | 0.907 | **0.905** | **0.944** | **0.978** | - | - |
| | ∃GNN-QE | **0.951** | 0.829 | 0.714 | 0.971 | **0.944** | 0.650 | 0.808 | **0.985** | **0.974** | 0.843 | 0.821 | 0.967 | 0.995 | 0.939 |
| mrr | GNN-QE | **0.885** | **0.693** | 0.587 | 0.797 | **0.835** | **0.704** | 0.699 | **0.447** | 0.417 | **0.420** | 0.301 | **0.343** | 0.741 | **0.610** |
| | ∃GNN-QE | 0.855 | 0.688 | 0.587 | **0.801** | 0.833 | 0.620 | **0.720** | 0.430 | **0.418** | 0.403 | **0.302** | 0.340 | **0.747** | 0.600 |

Table 2: Mean reciprocal rank and spearman rank correlation on test unanchored queries. Results of GNN-QE were obtained through our adaptation. Other metrics can be found in Appendix C.2

| Metric | Model | 1p | 2p | 3p | 2i | 3i | ip | pi | 2in | 3in | inp | pin | pni | 2u | up |
|---|---|---|---|---|---|---|---|---|---|---|---|---|---|---|---|
| | | | | | | No Anchor FB15k-237 | | | | | | | | | |
| spearmanr | GNN-QE | 0.629 | 0.899 | 0.865 | 0.966 | 0.970 | 0.932 | 0.894 | 0.811 | 0.802 | 0.651 | 0.711 | 0.893 | 0.553 | 0.937 |
| | ∃GNN-QE | **0.901** | **0.975** | **0.963** | **0.993** | **0.986** | **0.948** | **0.965** | **0.979** | **0.972** | **0.815** | **0.927** | **0.984** | **0.879** | 0.983 |
| mrr | GNN-QE | 0.007 | 0.081 | **0.087** | 0.221 | **0.295** | **0.108** | **0.164** | 0.020 | 0.105 | 0.063 | 0.036 | 0.022 | 0.029 | 0.078 |
| | ∃GNN-QE | **0.035** | **0.089** | 0.083 | **0.222** | 0.294 | 0.099 | 0.160 | **0.029** | 0.105 | **0.067** | **0.040** | **0.027** | **0.067** | **0.092** |
| | | | | | | No Anchor FB15k | | | | | | | | | |
| spearmanr | GNN-QE | 0.647 | 0.706 | 0.657 | 0.921 | 0.894 | 0.791 | 0.816 | 0.884 | 0.787 | 0.648 | 0.752 | 0.878 | 0.869 | 0.852 |
| | ∃GNN-QE | **0.924** | **0.789** | **0.660** | **0.957** | **0.959** | 0.539 | **0.803** | **0.964** | **0.971** | **0.804** | **0.824** | **0.967** | **0.967** | **0.897** |
| mrr | GNN-QE | 0.112 | 0.387 | 0.457 | **0.685** | 0.686 | 0.571 | 0.533 | 0.109 | 0.283 | 0.240 | 0.152 | 0.102 | 0.131 | 0.421 |
| | ∃GNN-QE | **0.228** | **0.434** | **0.496** | 0.613 | **0.718** | **0.579** | **0.572** | **0.190** | **0.325** | **0.269** | **0.189** | **0.158** | **0.292** | **0.455** |

The experiments show that training with unanchored queries (∃GNN-QE) results in a slight decrease in the mean reciprocal rank metric, and a slight increase in the spearman's rank correlation. We note that we failed to replicate the original numbers obtained in Zhu et al. (2022), so some of these differences may also be due to differences in training hardware. All in all, we see we are either paying a small price, or none at all, in order to enable a much more expressive class of queries. Note that the set of queries with best comparative performance is in queries with *negation*: this is according to our expectations, as negating a small set of entities results in dealing with large number of entities, just as in unanchored entry points.

**Tree-like queries.** Our second batch of results relates to enabling treatment of tree-like queries without anchors (**Q2**). While less interesting, we can also measure the effect of training with queries without anchors. In order to do this, we maintain weights computed by GNN-QE, but enable the processing of relation-projection that is non-anchored. Table 2 shows the results of both GNN-QE and ∃GNN-QE over the original test set of our benchmark databases. Results, as expected, suggest that training for this type of queries has a drastic increase in performance in all metrics.

**Cyclic Queries.** Next we move to cyclic queries, computed through their unravelings (**Q3**). Because our method relies on approximating the ground truth, we do not train for these types of queries, but rather try them directly in the trained models. To this extent, we construct a new test set for cyclic queries with 2 query-types: triangles, and squares (see Appendix B for more details).

Since our unravelings are parameterized by depth, before trying our all query shapes we tuned this parameter with an exploratory analysis for triangles and squares on FB15k-237. Here we are interested in maxing out the spearman correlation rank, because the choice of depth incides directly on the number of answers returned by each query. The result of this analysis for the triangle are shown in Figures 4 and 5. Further results can be found in Appendix C.

As we see, deeper unravelings appear to improve, but there seems to be a point after which this effect starts to be cancelled out by natural imprecisions of the trained model. This is evident when we analyze both the mean reciprocal rank (MRR) and the Spearman correlation. While the MRR tends to show slight improvements after a depth of 3, the Spearman correlation starts to diminish or worsen beyond that point. Hence, the remaining results (see Table 3) are reported for unravelings at depths 3 and 4 for triangles on all datasets. Notice that the metrics reported here are comparable to what state-of-the art architectures such as GNN-QE report for complex tree-like queries (see

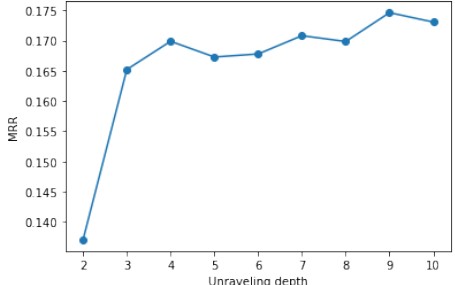
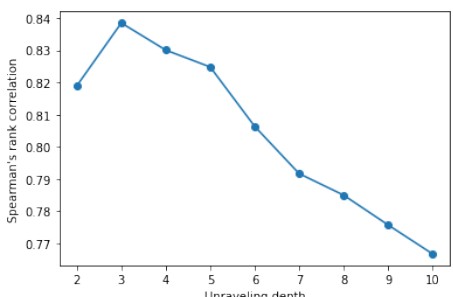

Figure 4: Mean reciprocal rank test results for different depths of unravelings for the triangle query.

Figure 5: Spearman rank correlation test results for different depths of unravelings for the triangle query.

Table 3: Spearman's rank correlation and reciprocal ranking scores of $\exists$GNN-QE model across unravelings of depths 3 and 4 for the triangle query.

|  |  |  | FB15k-237 | FB15k | NELL |
|---|---|---|---|---|---|
| Triangle | mrr | 3 | 0.138 | 0.354 | 0.207 |
|  |  | 4 | 0.136 | 0.359 | 0.206 |
|  | sp | 3 | 0.816 | 0.716 | 0.516 |
|  |  | 4 | 0.801 | 0.721 | 0.439 |

e.g. results for query type combining paths and intersection). We believe that our approximation scheme thus proves as a valid approach for allowing arbitrary CQs on neuro-symbolic architectures. Remaining results can be found on Appendix C.

## 6 FUTURE WORK

In this work, we present an approach to approximate the answers to arbitrary CQs over incomplete knowledge graphs by applying the mature toolbox developed for answering tree-like CQs. As for future work, we plan on expanding other neuro-symbolic architecture with the ability to deal with unanchored queries, so that we can also implement our approach in these architectures. While this approximation is cost-efficient, it can affect the quality of the retrieved answers. In fact, overapproximations may return answers that are not necessarily sound, even when the data is complete. One of our main goals for future work is to develop neuro-symbolic methods for CQs on knowledge graphs that return exact answers when evaluated on complete data. This process can be computationally demanding, but over the last decade, *worst-case optimal* algorithms have been developed for retrieving such answers in a symbolic manner (Ngo et al., 2013). We plan to investigate how such algorithms can be integrated into the neuro-symbolic framework studied in this paper to provide high-quality answers in a context where data is considered incomplete.

Another important issue we aim to address is determining the appropriate semantics for evaluating CQs over incomplete knowledge graphs. Neural approaches for completing knowledge graphs often produce a probability or score that indicates the likelihood of a link's existence between two given entities. This places us in the realm of *probabilistic* data. The data management community has long been studying how queries over probabilistic data should be interpreted (Suciu et al., 2011). We believe it is important to understand how this semantics aligns with the one used in the neuro-symbolic evaluation of tree-like CQs and how the techniques employed to approximate the probabilistic evaluation of CQs can be used in our setting.

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

## A MISSING DETAILS FROM SECTION 4

**Proposition A.1** (Proposition 4.1 in the body of the paper). *Let $q$ be a CQ and $d \geq 1$. The unraveling $\tilde{q}_d$ is a complete tree-like approximation of $q$. Moreover, $\tilde{q}_{d+1} \subseteq \tilde{q}_d$ holds.*

*Proof.* We first show that $\tilde{q}_d$ is a complete tree-like approximation of $q$. By definition, $\tilde{q}_d$ is tree-like and hence it suffices to show that $\tilde{q}_d$ contains $q$, that is, that there exists a homomorphism $h$ from $\tilde{q}_d$ to $q$. Consider the mapping $h : \mathsf{Var}(\tilde{q}_d) \cup \mathsf{Con}(\tilde{q}_d) \rightarrow \mathsf{Var}(q) \cup \mathsf{Con}(q)$ such that $h(a) = a$ for all $a \in \mathsf{Con}(\tilde{q}_d)$ and $h(z_P) = \mathsf{end}(P) \in \mathsf{Var}(q)$ for all $z_P \in \mathsf{Var}(\tilde{q}_d)$. We claim that $h$ is a homomorphism. Take an atom $R(o_P, o_{P'})$ in $\tilde{q}_d$. By construction, $R(\mathsf{end}(P), \mathsf{end}(P'))$ is an atom in $q$. For any valid path $W$, $\mathsf{end}(W) = h(o_W)$. Indeed, if $W$ is unanchored, then $h(o_W) = h(z_W) = \mathsf{end}(W)$. If $W$ is anchored, $h(o_W) = h(\mathsf{end}(W)) = \mathsf{end}(W)$. It follows that $R(h(o_P), h(o_{P'}))$ is an atom in $q$ as required.

For the second part of the proposition, note that, by construction, the atoms of $\tilde{q}_d$ are contained in the atoms of $\tilde{q}_{d+1}$. In particular, the identity mapping from $\mathsf{Var}(\tilde{q}_d) \cup \mathsf{Con}(\tilde{q}_d)$ to $\mathsf{Var}(\tilde{q}_{d+1}) \cup \mathsf{Con}(\tilde{q}_{d+1})$ is a homomorphism from $\tilde{q}_d$ to $\tilde{q}_{d+1}$. It follows that $\tilde{q}_{d+1} \subseteq \tilde{q}_d$. □

**Theorem A.1** (Theorem 4.1 in the body of the paper). *Let $q$ be a CQ and $d \geq 1$. Suppose $q'$ is a complete tree-like approximation of depth at most $d$. Then $\tilde{q}_d \subseteq q'$ holds.*

*Proof.* We define a *path* of $q(x)$ as a sequence $x_0, A_1, x_1, \ldots, A_k, x_k$, for $k \geq 0$, such that:

- $x_0 = x$, each $x_i \in \mathsf{Var}(q) \cup \mathsf{Con}(q)$, and each $A_i$ is an atom of $q$.
- for each $1 \leq i \leq k$, the atom $A_i$ is either of the form $R(x_{i-1}, x_i)$ (a forward traversal of the atom), or $R(x_i, x_{i-1})$ (a backward traversal of the atom).

In other words, a path is a valid path without the validity condition on consecutive atoms. As for valid paths, we denote by $\mathsf{end}(P)$ the element at the end of the path $P$. Note that every path $P$ that is not valid define a unique valid path $\mathsf{valid}(P)$ such that $\mathsf{end}(P) = \mathsf{end}(\mathsf{valid}(P))$. Indeed, whenever we have a subsequence in the path $P$ of form $y, A, z, A, y$ violating validity, we can replace it by $y$. By iteratively, applying this simplification, we always obtain a unique valid path $\mathsf{valid}(P)$ with $\mathsf{end}(P) = \mathsf{end}(\mathsf{valid}(P))$.

Now suppose $q'(x)$ is a complete tree-like approximation of $q(x)$ of depth at most $d$. In particular, there exists a homomorphism $h$ from $q'$ to $q$. We shall define a homomorphism $g$ from $q'$ to $\tilde{q}_d$,

Table 4: Statistic of unanchored query set for each dataset: FB15k-237, FB15k and NELL.

| | Split | 1p | 2p | 3p | 2i | 3i | 2in | 3in | inp | pin | pni | ip | pi | 2u | up |
|---|---|---|---|---|---|---|---|---|---|---|---|---|---|---|---|
| **FB15k-237** | train | 474 | 13139 | 62826 | 117688 | 147721 | 11644 | 12790 | 5719 | 12286 | 13358 | - | - | - | - |
| | valid | 288 | 2514 | 4213 | 3763 | 3368 | 2272 | 4255 | 3330 | 4080 | 1970 | 4135 | 4396 | 2905 | 1613 |
| | test | 295 | 2475 | 4234 | 3792 | 3471 | 2259 | 4219 | 3272 | 3941 | 1975 | 3903 | 4312 | 2901 | 1604 |
| **FB15k** | train | 2690 | 51378 | 172943 | 231273 | 271760 | 22250 | 23759 | 11016 | 23337 | 25222 | - | - | - | - |
| | valid | 1182 | 5246 | 7481 | 6499 | 5072 | 3592 | 6908 | 5825 | 6506 | 3173 | 6620 | 7093 | 4916 | 2559 |
| | test | 1240 | 5302 | 7433 | 6527 | 5252 | 3558 | 6902 | 5815 | 6442 | 2979 | 6306 | 7125 | 4906 | 2565 |
| **NELL** | train | 400 | 8713 | 35045 | 50076 | 37010 | 4458 | 3015 | 1035 | 4218 | 5026 | - | - | - | - |
| | valid | 346 | 2118 | 3239 | 2731 | 2342 | 1721 | 3193 | 2643 | 2884 | 1410 | 2772 | 3421 | 2428 | 1249 |
| | test | 342 | 2050 | 3192 | 1596 | 897 | 474 | 1725 | 1568 | 1097 | 373 | 1542 | 1841 | 922 | 241 |

which would imply $\tilde{q}_d \subseteq q'$ as required. For $w \in \mathsf{Var}(q')$, we define the path $P_w$ in $q$ as follows. Take the unique path from the root $x$ to $w$ in $q'$. As $h$ is a homomorphism, the image of this path via $h$ produces a path in $q$, which we denote $P_w$. Consider the mapping $g$ from $\mathsf{Var}(q') \cup \mathsf{Con}(q')$ to $\mathsf{Var}(\tilde{q}_d) \cup \mathsf{Con}(\tilde{q}_d)$ such that $g(a) = a$ for all $a \in \mathsf{Con}(q')$, and $g(w) = o_{\mathsf{valid}(P_w)}$ for all $w \in \mathsf{Var}(q')$. Recall that $o_W = z_W$ if $W$ is a unanchored valid path and $o_W = \mathsf{end}(W)$ otherwise. Note $g$ is well-defined: as the depth of $q'$ is $\leq d$, then the lengths of the paths $P_w$, and hence $\mathsf{valid}(P_w)$, are always $\leq d$. We claim that $g$ is a homomorphism. Take an atom $R(t, t')$ in $q'$ and suppose $t$ is the parent of $t'$ in $q'$ (the other case is analogous). We consider some cases:

- $t, t' \in \mathsf{Var}(q')$: Note that either $\mathsf{valid}(P_{t'})$ extends $\mathsf{valid}(P_t)$, that is, $\mathsf{valid}(P_{t'}) = \mathsf{valid}(P_t), A', w'$, or $\mathsf{valid}(P_t)$ extends $\mathsf{valid}(P_{t'})$. Indeed, if the last atom $A'$ of $P_{t'}$ is different from the last atom of $\mathsf{valid}(P_t)$ then $\mathsf{valid}(P_{t'}) = \mathsf{valid}(P_t), A, w'$. Otherwise, if the last atom $A'$ of $P_{t'}$ coincides with the last atom of $\mathsf{valid}(P_t)$, then $\mathsf{valid}(P_{t'})$ is the subsequence of $\mathsf{valid}(P_t)$ that ends just before traversing the last atom $A'$. In particular, $\mathsf{valid}(P_t) = \mathsf{valid}(P_{t'}), A', w$. Suppose $\mathsf{valid}(P_{t'}) = \mathsf{valid}(P_t), A', w'$ (the other case is analogous). Note that $A' = R(h(t), h(t')) = R(\mathsf{end}(\mathsf{valid}(P_t)), \mathsf{end}(\mathsf{valid}(P_{t'})))$ and $w' = h(t')$. By construction, the atom $R(o_{\mathsf{valid}(P_t)}, o_{\mathsf{valid}(P_{t'})})$ is in $\tilde{q}_d$. It follows that $R(g(t), g(t'))$ belongs to $\tilde{q}_d$ as required.

- $t \in \mathsf{Con}(q')$ or $t' \in \mathsf{Con}(q')$: Suppose $t \in \mathsf{Var}(q')$ and $t' \in \mathsf{Con}(q')$ (the other cases are analogous). Let $P'$ be the path in $q'$ moving from the root $x$ to $t$ and then going through the atom $R(t, t')$ and ending at the constant $t'$. Let $P$ be the path in $q$ that is the image via $h$ of the path $P'$ in $q'$. We can repeat the argument above by replacing $P_{t'}$ with $P$.

$\square$

# B  EXPERIMENTAL DETAILS

## B.1  NEW QUERY SET

We use both the query set provided by Ren & Leskovec (2020) and a new set of general tree like queries. Statistic of the new query set are provided in Table 4.

For testing our approximation scheme, we also generated 500 triangles and 500 squares on FB15k-237 and NELL and their corresponding unravelings.

## B.2  TRAINING HYPERPARAMETERS

The training hyperparameters utilized in this study closely follow those detailed in the GNN-QE paper (Zhu et al. (2022)). However, in order to accommodate the specific hardware constraints of our computational setup, we made adjustments to the batch size while keeping all other hyperparameters consistent. Table 5 show the learning hyperparameters we used[3].

It's important to note that while working with the NELL dataset, we ran into hardware limitations that unfortunately made it challenging to completely replicate the results outlined in the original

---

[3]To ensure a fair comparison, we maintained the architecture hyperparameters, including the number of layers and hidden dimensions, as specified in the original GNN-QE paper.

Table 5: Learning hyperparameters of ∃GNN-QE for FB15k237, FB15k and NELL.

| Hyperparameter | FB15k-237 | FB15k | NELL |
|---|---|---|---|
| Batch size | 24 | 24 | 6 |
| Optimizer | Adam | Adam | Adam |
| Learning rate | 5e-3 | 5e-3 | 5e-3 |
| Adv. temperature | 0.2 | 0.2 | 0.2 |
| #Batches | 10.000 | 10.000 | 18.000 |

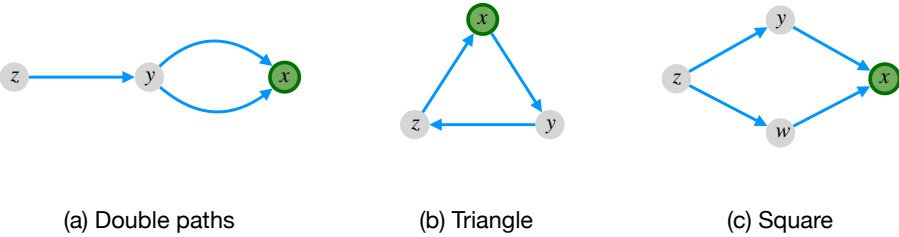

(a) Double paths      (b) Triangle      (c) Square

Figure 6: Cyclic queries used for testing the method. Double paths where unravelled up to depth 2 and 3, triangles where unravelled up to depth 3 and 4, and squares where unravelled up to depths 4 and 6.

GNN-QE paper. A comparison of the results we replicated and the ones reported in the paper can be found in Table 7.

# C  ADDITIONAL EXPERIMENTAL RESULTS

## C.1  ANCHORED TREE-LIKE

Table 6 presents H@1 and MAPE for FB15k-237 and FB15k.

Table 7 presents H@1, MRR, MAPE and Spearman's rank correlation for NELL. Due to different training hyperparameters, we report both the metrics showed in GNN-QE paper Zhu et al. (2022) and GNN-QE trained under the parameters in Table 5

## C.2  QUERIES WITHOUT ANCHOR

Table 8 shows H@1, MAPE and MRR on test unanchored queries for both GNN-QE and ∃GNN-QE

## C.3  CYCLIC CONJUNCTIVE QUERIES

Figures 7 and 8 show the MRR and spearman rank correlation for squares using unravelings of differents depths.

Tables 9 and 10 show H@1, MAPE and MRR for different unravelings for both triangles and squares respectively.

Table 6: Hits@1 and MAPE of test evaluation over FB15k-237 and FB15k. Metrics of GNN-QE are retrieved from its original paper.

| Metric | Model | 1p | 2p | 3p | 2i | 3i | ip | pi | 2in | 3in | inp | pin | pni | 2u | up |
|--------|-------|----|----|----|----|----|----|----|-----|-----|-----|-----|-----|----|----|
| | | | | | | | **FB15k-237** | | | | | | | | |
| hits@1 | GNN-QE | 0.328 | 0.082 | 0.065 | 0.277 | 0.446 | 0.123 | 0.224 | 0.041 | 0.081 | 0.041 | 0.025 | 0.027 | 0.098 | 0.076 |
| | ∃GNN-QE | 0.228 | 0.056 | 0.051 | 0.231 | 0.401 | 0.095 | 0.187 | 0.022 | 0.058 | 0.034 | 0.016 | 0.014 | 0.067 | 0.054 |
| mape | GNN-QE | 0.409 | 0.236 | 0.274 | 0.348 | 0.534 | 0.600 | 0.399 | 0.403 | 0.526 | 0.496 | 0.448 | 0.290 | 0.278 | 0.203 |
| | ∃GNN-QE | 0.425 | 0.249 | 0.334 | 0.383 | 0.573 | 0.414 | 0.410 | 0.408 | 0.512 | 0.463 | 0.468 | 0.333 | 0.314 | 0.292 |
| | | | | | | | **FB15k** | | | | | | | | |
| hits@1 | GNN-QE | 0.861 | 0.635 | 0.525 | 0.748 | 0.801 | 0.651 | 0.636 | 0.354 | 0.331 | 0.338 | 0.186 | 0.218 | 0.671 | 0.530 |
| | ∃GNN-QE | 0.831 | 0.643 | 0.533 | 0.759 | 0.791 | 0.566 | 0.668 | 0.308 | 0.297 | 0.315 | 0.189 | 0.219 | 0.704 | 0.540 |
| mape | GNN-QE | 0.344 | 0.297 | 0.347 | 0.391 | 0.573 | 0.346 | 0.478 | 0.314 | 0.503 | 0.503 | 0.394 | 0.298 | 0.135 | 0.265 |
| | ∃GNN-QE | 0.264 | 0.293 | 0.340 | 0.367 | 0.540 | 0.386 | 0.453 | 0.346 | 0.480 | 0.457 | 0.426 | 0.292 | 0.106 | 0.239 |

Table 7: Spearman's rank correlation and MRR of test queries over NELL dataset (the results under GNN-QE corresponds to the ones in GNN-QE's paper. GNN-QE* denotes the results using training hyperparameters of Table 5)

| Metric | Model | 1p | 2p | 3p | 2i | 3i | ip | pi | 2in | 3in | inp | pin | pni | 2u | up |
|--------|-------|----|----|----|----|----|----|----|-----|-----|-----|-----|-----|----|----|
| | | | | | | | **NELL** | | | | | | | | |
| spearmanr | GNN-QE | 0.913 | 0.851 | 0.780 | 0.974 | 0.935 | 0.737 | 0.825 | 0.994 | 0.980 | 0.882 | 0.848 | 0.976 | - | - |
| | GNN-QE(*) | 0.942 | 0.843 | 0.752 | 0.972 | 0.942 | 0.725 | 0.812 | 0.988 | 0.974 | 0.828 | 0.813 | 0.971 | 0.991 | 0.938 |
| | ∃GNN-QE | 0.951 | 0.829 | 0.714 | 0.971 | 0.944 | 0.650 | 0.801 | 0.985 | 0.974 | 0.843 | 0.821 | 0.967 | 0.995 | 0.939 |
| mrr | GNN-QE | 0.533 | 0.189 | 0.149 | 0.424 | 0.525 | 0.189 | 0.308 | 0.159 | 0.126 | 0.099 | 0.146 | 0.114 | 0.063 | 0.063 |
| | GNN-QE(*) | 0.492 | 0.175 | 0.138 | 0.394 | 0.499 | 0.169 | 0.284 | 0.086 | 0.131 | 0.111 | 0.054 | 0.052 | 0.150 | 0.116 |
| | ∃GNN-QE | 0.479 | 0.160 | 0.119 | 0.378 | 0.484 | 0.140 | 0.269 | 0.086 | 0.128 | 0.111 | 0.055 | 0.053 | 0.141 | 0.107 |

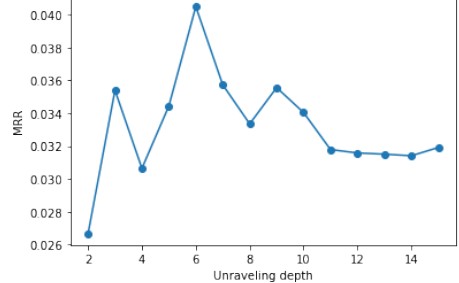

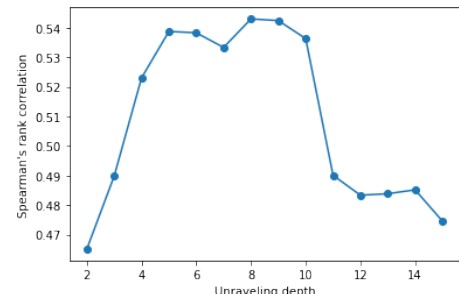

Figure 7: MRR versus depth of square unravelings.

Figure 8: Spearman rank correlation versus depth of square unravelings.

Table 8: Results of evaluation of GNN-QE and ∃GNN-QE over test set of unanchored queries. GNN-QE metrics are retrived from our training of GNN-QE following hyperparameters in Table 5

| Metric | Model | 1p | 2p | 3p | 2i | 3i | ip | pi | 2in | 3in | inp | pin | pni | 2u | up |
|--------|-------|----|----|----|----|----|----|----|-----|-----|-----|-----|-----|----|----|
| | | | | | | | **FB15k-237** | | | | | | | | |
| hits@1 | GNN-QE | 0.001 | 0.044 | 0.045 | 0.151 | 0.205 | 0.064 | 0.105 | 0.004 | 0.048 | 0.025 | 0.011 | 0.005 | 0.009 | 0.044 |
| | ∃GNN-QE | 0.015 | 0.045 | 0.042 | 0.143 | 0.196 | 0.055 | 0.100 | 0.008 | 0.041 | 0.027 | 0.011 | 0.006 | 0.034 | 0.051 |
| mape | GNN-QE | 3.22 | 0.340 | 0.332 | 0.489 | 0.431 | 0.366 | 0.454 | 0.763 | 0.652 | 0.681 | 0.623 | 0.734 | 2.329 | 0.263 |
| | ∃GNN-QE | 0.626 | 0.178 | 0.199 | 0.245 | 0.333 | 0.307 | 0.291 | 0.440 | 0.514 | 0.501 | 0.496 | 0.347 | 0.793 | 0.159 |
| | | | | | | | **FB15k** | | | | | | | | |
| hits@1 | GNN-QE | 0.075 | 0.315 | 0.387 | 0.552 | 0.620 | 0.511 | 0.470 | 0.054 | 0.178 | 0.162 | 0.074 | 0.050 | 0.084 | 0.347 |
| | ∃GNN-QE | 0.171 | 0.353 | 0.426 | 0.628 | 0.661 | 0.523 | 0.511 | 0.103 | 0.209 | 0.181 | 0.094 | 0.081 | 0.209 | 0.373 |
| mape | GNN-QE | 11.18 | 0.488 | 0.517 | 0.459 | 0.547 | 0.478 | 0.551 | 0.778 | 0.604 | 0.638 | 0.533 | 0.645 | 0.926 | 0.383 |
| | ∃GNN-QE | 5.99 | 0.29 | 0.306 | 0.280 | 0.395 | 0.351 | 0.370 | 0.447 | 0.453 | 0.454 | 0.442 | 0.317 | 0.361 | 0.205 |
| | | | | | | | **NELL** | | | | | | | | |
| hits@1 | GNN-QE | 0.004 | 0.022 | 0.033 | 0.130 | 0.183 | 0.057 | 0.082 | 0.066 | 0.021 | 0.025 | 0.004 | 0.004 | 0.005 | 0.017 |
| | ∃GNN-QE | 0.010 | 0.029 | 0.038 | 0.142 | 0.190 | 0.056 | 0.084 | 0.097 | 0.022 | 0.027 | 0.006 | 0.006 | 0.012 | 0.012 |
| mape | GNN-QE | 2.698 | 0.571 | 0.609 | 0.518 | 0.641 | 0.583 | 0.554 | 0.689 | 0.635 | 0.697 | 0.594 | 0.825 | 1.328 | 0.345 |
| | ∃GNN-QE | 0.325 | 0.428 | 0.524 | 0.461 | 0.581 | 0.476 | 0.478 | 0.314 | 0.458 | 0.534 | 0.432 | 0.287 | 0.213 | 0.302 |
| mrr | GNN-QE | 0.007 | 0.052 | 0.069 | 0.211 | 0.276 | 0.110 | 0.139 | 0.023 | 0.075 | 0.059 | 0.025 | 0.018 | 0.012 | 0.043 |
| | ∃GNN-QE | 0.028 | 0.067 | 0.075 | 0.228 | 0.290 | 0.107 | 0.137 | 0.034 | 0.080 | 0.066 | 0.033 | 0.024 | 0.033 | 0.047 |
| spearmanr | GNN-QE | 0.589 | 0.732 | 0.591 | 0.928 | 0.922 | 0.520 | 0.731 | 0.922 | 0.833 | 0.655 | 0.646 | 0.907 | 0.735 | 0.913 |
| | ∃GNN-QE | 0.924 | 0.789 | 0.660 | 0.957 | 0.959 | 0.539 | 0.803 | 0.964 | 0.971 | 0.804 | 0.824 | 0.967 | 0.967 | 0.897 |

Table 9: Hits@1 and MAPE test results for triangle's unravelings of depth 2 to 10 over FB15k-237.

| Depth | 2 | 3 | 4 | 5 | 6 | 7 | 8 | 9 | 10 |
|-------|---|---|---|---|---|---|---|---|----|
| hits@1 | 0.060 | 0.078 | 0.068 | 0.066 | 0.076 | 0.073 | 0.072 | 0.076 | 0.073 |
| mape | 2.148 | 1.134 | 0.914 | 0.789 | 0.748 | 0.734 | 0.724 | 0.710 | 0.711 |

Table 10: Hits@1 and MAPE test results for square's unravelings of depth 2 to 14 over FB15k-237.

| Depth | 2 | 3 | 4 | 5 | 6 | 7 | 8 | 9 | 10 | 11 | 12 | 13 | 14 |
|-------|---|---|---|---|---|---|---|---|----|----|----|----|----|
| hits@1 | 0.005 | 0.010 | 0.005 | 0.010 | 0.014 | 0.010 | 0.010 | 0.010 | 0.010 | 0.010 | 0.010 | 0.010 | 0.010 |
| mape | 5.364 | 3.837 | 3.377 | 3.053 | 2.923 | 2.849 | 2.675 | 2.637 | 2.684 | 2.845 | 2.840 | 2.834 | 2.834 |

