# OpenReview forum: "A neuro-symbolic framework for answering conjunctive queries"
_ICLR.cc/2024/Conference — Submitted to ICLR 2024_

### Official Review · Reviewer_tybU · 2023-10-13

**Soundness:** 2 fair
**Presentation:** 3 good
**Contribution:** 2 fair
**Rating:** 3
**Confidence:** 5

**Summary:**

This paper addresses the challenge of answering logical queries over incomplete knowledge graphs (KGs). The authors argue that current approaches are limited in that they focus on monadic anchored tree-shaped queries, characterized by query dependency graphs with a tree structure and constant values as leaf nodes. To address this limitation, the paper claims the following contributions:

(C1) A technique for over-approximating arbitrary monadic CQs as tree-shaped CQs (without anchors). This means that, given a CQ q, the technique would generate a tree shaped q’ that subsumes q (meaning that each answer to q on any dataset is contained in the answer set for q’ on the same dataset) and which is “optimal” in some well-defined way

(C2) A proposal to adapt the method of Zhu et al. for anchored tree-shaped CQs to the unanchored setting

Additionally, the authors provide empirical results based on established benchmarks related to these tasks.

**Strengths:**

The topic of query answering over incomplete KGs has attracted significant attention in recent years. Hence, this submission is clearly relevant to ICLR. Furthermore, the problem of approximating CQs has also received attention within the database theory and knowledge representation research communities. The paper is also clearly written and the main formal claims in the paper appear correct. The paper does make a contribution to the current state-of-the-art, albeit one that I consider rather modest (for specific elaboration, please see below).

**Weaknesses:**

As previously mentioned, I find the claimed contribution rather limited. Specifically, Contribution (C1) is not directly related to the field of Machine Learning; the results it claims are strictly within the domain of database theory. Unravelling techniques are conventional in database theory and knowledge representation. Furthermore, the observation that the tree unraveling (to any depth) of a non-tree-shaped conjunctive query results in an over-approximation of the original query might be considered common knowledge within the community. Therefore, as a researcher with main background in database theory, logic, and knowledge representation, I regard the primary findings presented in Section 4.1 as straightforward observations that might not carry sufficient significance for publication.

The significance of (C1) in the context of the current machine learning-based query answering state-of-the-art is also not evident to me. As the authors have mentioned, existing approaches are unable to handle arbitrary monadic tree-shaped conjunctive queries (CQs) without the requirement of query anchoring. Consequently, most of these approaches cannot directly leverage the proposed approximation. This brings us to Contribution (C2), in which the authors introduce an expansion of Zhu et al.'s method to encompass unanchored queries. This extension is not sufficiently elaborated and it is unclear it what manner it enables the technique by Zhu et al to "support" arbitrary tree shaped CQs. Additionally, it is uncertain whether this extension can be applied to other methods, especially those dependent on the existence of embeddings for the anchors.  This doesn't seem straightforward in my view.

I also found the experimental results somewhat perplexing. Specifically, in the comparison between GNN-QE and its extension, \exists GNN-QE, applied to anchored queries, both systems are trained on distinct datasets; the significance of the reported results remains unclear to me in this context. Adding to the confusion, the experiments appear to include results for queries that may not strictly adhere to conjunctive queries (CQs) and may incorporate other first-order constructs, including negation. To the best of my knowledge, the results presented in the core technical sections of the paper are confined to CQs (and indeed, the seminal result by Chandra and Merlin only applies to CQs).

**Questions:**

- Please clarify the applicability of your results to queries involving disjunction and negation.

- Please clarify whether methods other than that by Zhu et al. can be easily extended to support CQs.

---

> ### Author Response · Authors · 2023-11-21
>
> Thank you for you review and feedback. We address your concerns below.
>
> > Contribution (C1) is not directly related to the field of Machine Learning; the results it claims are strictly within the domain of database theory...
>
> Please note that we are not claiming the notion of tree-like approximation itself as a contribution, and we mention it has been studied before in the second paragraph of Section 2. We agree with the reviewer that unravelings are common in the literature and that the completeness claim is straightforward. The optimality claim is less straightforward, but it is not difficult. We are not aware of any optimality result of this kind (to the best of our knowledge, similar previous results deal with treewidth-1 queries, as mentioned in Section 2). If the reviewer has a concrete reference showing a similar result, we are happy to add it to the paper. Also, we are happy to improve the wording so it is clear to the reader that we are not claiming the notion of tree-like approximation as a contribution.
>
> The contribution, as explained in the paper, is the use of tree-like approximations of conjunctive queries in the context of Machine Learning, to obtain neuro-symbolic models that can handle cyclic conjunctive queries. We stress that handling cyclic queries has been identified as an open problem in the literature (Ren et al 2023).
>
> > The significance of (C1) in the context of the current machine learning-based query answering state-of-the-art is also not evident to me. As the authors have mentioned, existing approaches are unable to handle arbitrary monadic tree-shaped conjunctive queries (CQs) without the requirement of query anchoring...
>
> We are not claiming (C1) as a contribution, see our previous answer. As far as we can see, it is not clear how to define a notion of tree-like approximations that can be handled directly by current methods and that enjoys strong theoretical guarantees (e.g. completeness, optimality). On the other hand, we believe that the fact that current methods need to be adapted to use tree-like approximations, does not make our contribution less significant.
>
> > ...it is unclear in what manner it enables the technique by Zhu et al to "support" arbitrary tree shaped CQs.
>
> We describe this in the 3rd paragraph of Section 4.3. The idea is that GNN-QE (Zhu et al) is a method that allows one to encompass existential quantification in the latent space as a full unitary vector (in other more complex methods one would probably have to devise a more clever encoding scheme to allow for unanchored queries).  We would appreciate any concrete suggestions for improving the presentation.
>
> > I also found the experimental results somewhat perplexing. Specifically, in the comparison between GNN-QE and its extension, \exists GNN-QE, applied to anchored queries, both systems are trained on distinct datasets; the significance of the reported results remains unclear to me...  Adding to the confusion, the experiments appear to include results for queries that may not strictly adhere to conjunctive queries (CQs) and may incorporate other first-order constructs, including negation. To the best of my knowledge, the results presented in the core technical sections of the paper are confined to CQs (and indeed, the seminal result by Chandra and Merlin only applies to CQs).
> > Questions:
> Please clarify the applicability of your results to queries involving disjunction and negation.
>
> Both GNN-QE and \exists GNN-QE are trained on the same knowledge graphs, but different set of queries: GNN-QE is trained on tree-like anchored queries and \exists GNN-QE is trained on tree-like queries (with and without anchors). In other words, the training\validation set of GNN-QE is a proper subset of the training\validation set of \exists GNN-QE. Experiments show that training on unanchored tree-like queries improves the performance of the model in such queries without compromising its original objective (anchored tree-like queries): \exists GNN-QE achieves competitive results in both anchored and unanchored tree-like queries.
>
> Regarding negation, our approximation scheme is indeed devised only for CQs, but GNN-QE and \exists GNN-QE are architectures capable of answering arbitrary anchored and arbitrary tree-like queries, which may include union and negation. Hence we include two different type of tests: we compare GNN-QE and \exists GNN-QE in terms of their ability to answer tree-like queries, and then test how  \exists GNN-QE handles (cyclic) CQs via our approximation scheme.
>
> > Please clarify whether methods other than that by Zhu et al. can be easily extended to support CQs.
>
> We agree with the reviewer. It is not clear how to adapt other methods to handle existential variables. This is why in this paper we focus on GNN-QE which naturally admits this extension. We believe that trying to adapt other methods (or propose new methods) that handle existential variables is an interesting question for future research.

---

> > ### Comment · Reviewer_tybU · 2023-11-21
> >
> > I thank the authors for their response. The contribution seems clear to me, but I still feel that it is insufficient for ICLR. I encourage the authors to continue working in this direction and develop their results further.
> > Concerning experiments, I understand that the underpinning KGs are the same. However, models are trained on different sets of queries, and hence, unless I am mistaken, the training sets are different for each model. This could potentially compromise the validity of the experimental results.

---

> > > ### Author Response · Authors · 2023-11-21
> > > **Response to reviewer**
> > >
> > > Thank you for the comments.
> > >
> > > > The contribution seems clear to me, but I still feel that it is insufficient for ICLR. I encourage the authors to continue working in this direction and develop their results further.
> > >
> > > Thanks for the support. We believe that our method **considerably extends the classes of queries answerable** (or at least approximately answerable) using  **sota neuro-symbolic methods**. Evaluating general queries in general requires the development of efficient higher-order tensor methods which are currently out of reach in practice.
> > >
> > > > However, models are trained on different sets of queries, and hence, unless I am mistaken, the training sets are different for each model. This could potentially compromise the validity of the experimental results.
> > >
> > > They are indeed trained with different sets of queries: in fact the training\validation set of GNN-QE is a proper subset of the training\validation set of \exists GNN-QE. The point in making the comparisons between both was to show that training on unanchored tree-like queries improves the performance of the model in such queries without compromising its original objective (anchored tree-like queries).

---

### Official Review · Reviewer_vTgu · 2023-10-24

**Soundness:** 1 poor
**Presentation:** 1 poor
**Contribution:** 1 poor
**Rating:** 3
**Confidence:** 2

**Summary:**

The authors propose a novel neuro-symbolic framework for approximating complex queries on knowledge graphs. The method uses tree-like queries to approximate complex conjunctive queries and is implemented on top of GNN-QE. Some experiment results on FB15K, FB15k-237, and Nell995 datasets outperform SOTA level.

**Strengths:**

not identified yet.

**Weaknesses:**

The presentation of the paper is poor. This prevents the understanding of the content. The motivation and the research question are not clear. The experiment results are not always better than the based-line. But, only experiments on benchmark datasets FB15K, FB15k-237 are not sufficient to support authors' second and third contribution claims.

**Questions:**

Why shall we be interested in the research of answering arbitrary conjunctive queries over incomplete knowledge graphs?

Would this method also work for complete knowledge graphs?

What is the intuition behind the idea of "approximating a cyclic query by an infinite family of tree-like queries"?

What if a relation is self-reflective?

What do you mean by "neuro-symbolic framework"?

---

> ### Author Response · Authors · 2023-11-21
>
> We regret that the reviewer finds the presentation poor and we will do our utmost best to improve the presentation. Any specific suggestions by the reviewer as to where things can be improved would be appreciated.
>
> > The motivation and the research question are not clear.
>
> **Motivation:** Current state of the art only supports tree-like queries, how to use this to deal with more complex queries? **Answer:** tree-like approximations. **Research question:** What are such approximations and how do these work in practice? We answer both questions in the paper.
>
> > The experiment results are not always better than the based-line.
>
> Unfortunately we were not able to fully reproduce results from the original GNN-QE paper due to hardware limitations.
>
> In the case of anchored tree-like queries, on our implementation and training, results are closer, while still being slightly better in the original GNN-QE. We believe that this might be a result of the way we created unanchored queries.
>
> In the case of unanchored tree-like queries, the results are almost always better for \exists GNN-QE, and in the cases where GNN-QE is better the difference is very small.
>
> > But, only experiments on benchmark datasets FB15K, FB15k-237 are not sufficient to support authors' second and third contribution claims.
>
> We also report on NELL in the supplementary material as mentioned in the paper. This is precisely the same experimental setup presented in the original GNN-QE paper and other previous models such as Q2B, betaE and conE.
>
> > Questions:
> Why shall we be interested in the research of answering arbitrary conjunctive queries over incomplete knowledge graphs?
>
> Knowledge bases, and graph databases in general, are one of the most important trends in data management, with dozens of engines available, large public graphs such as Wikidata, and interest fro big companies such as Apache, Google or Amazon.
>
> Answering conjunctive queries in knowledge graphs is important because they serve as the building block of most knowledge graph query languages (see e.g. Hogan, et al. "Knowledge graphs." ACM Computing Surveys 2021). Conjunctive queries are also important outside knowledge graphs, as they serve as a formalization of SELECT FROM WHERE statements in SQL.
>
> Finally, since knowledge graphs are commonly used to capture information, one can rarely assume knowledge graphs are complete. The most prominent example is Wikidata, serving as the knowledge graph of Wikipedia: since the aim of Wikipedia is to capture all information known to the human world, in practice we will never have a complete version of it. Hence, link prediction is important in knowledge graphs, and so is being able to answer queries (specially conjunctive queries) in the presence of missing links.
>
> > Would this method also work for complete knowledge graphs?
>
> The model does not incorporate a way of letting it know the input graph is complete, so even if the graph is complete, it will continue trying to predict the most likely links, and using them to answer queries. However, in training we will see all relations present in the graph, so the error should be minimal.
> Although knowledge graphs in practice, as we mention, are usually incomplete, if, theoretically speaking, we find a complete knowledge graph, it will be more efficient to use the standard database machinery developed for complete knowledge graphs, as it is specifically designed for this scenario.
>
> > What is the intuition behind the idea of "approximating a cyclic query by an infinite family of tree-like queries"?
>
> The idea is to have tree-like queries that approximate the original query in the sense that their answer sets are close, independently of the underlying knowledge graph. As we take better approximations in the family we expect their answer sets to be closer to the answer set of the original query.
>
> > What if a relation is self-reflective?
>
> Our approach is oblivious to the fact that a knowledge graph relation may be self-reflective. If you mean the presence of self-loops in the conjunctive queries, these are considered as cycles. Our approximation approach still applies in the presence of self-loops.
>
> > What do you mean by "neuro-symbolic framework"?
>
> We mean methods that work with neural architectures, but in which we introduce a specific bias, obliging that the query processing algorithm uses the symbolic information in how the query is written in a formal query language.

---

> > ### Comment · Reviewer_vTgu · 2023-11-22
> >
> > If your "neuro-symbolic framework" is to "introduce a specific bias, obliging that the query processing algorithm uses the symbolic information in how the query is written in a formal query language", your method is only an alchemy, and is not on the right track to approach to (and finally reach) the performance of symbolic logic query.

---

### Official Review · Reviewer_Ajos · 2023-11-01

**Soundness:** 3 good
**Presentation:** 3 good
**Contribution:** 3 good
**Rating:** 6
**Confidence:** 2

**Summary:**

The paper proposes a framework for answering arbitrary conjunctive queries over incomplete knowledge graphs. The main idea of the approach is to approximate a cycle query by an infinite family of tree-like queries, and leverage existing models for the latter. Such approximations come with strong guarantees, namely completeness and optimality.

**Strengths:**

- The paper was, for the most paper, well-written and easy to follow.
- In a neuro-symbolic setting, the authors are the first to tackle the problem of answering cyclic queries on incomplete knowledge graphs.
- The proposed approach is quite intuitive and simple, essentially a linear approximation of the logical query. This has the added benefit that, once approximated, the task of answering the logical query can be delegated to any state-of-the-art near-symbolic query processor.
- The approximation is guaranteed to be complete, as well as optimal for a given computational budget.

**Weaknesses:**

- One apparent weakness seems to be the addition of yet another hyper-parameter $d$ which determines the depth of the tree to which the cyclic logical query is unraveled.

- The proposed approach seems to achieve a lower performance compared to the baseline when evaluated on anchored tree-like queries

**Questions:**

- Do you have any intuition as to why the proposed approach seems to perform worse, on average, compared to the baseline on anchored tree-like queries?

- In the experimental setup you mentioned that you "additionally provide a new set of training, validation and test queries...". Is this in addition to the unanchored set originally in the dataset? I was under the impression that your method could only handle unanchored queries?

---

> ### Author Response · Authors · 2023-11-21
>
> Thank you for your comments. We address your questions below:
>
> > Weaknesses:
> > One apparent weakness seems to be the addition of yet another hyper-parameter which determines the depth of the tree to which the cyclic logical query is unraveled.
>
> We agree, and this is the reason we did an empirical fitting of the depth. Experiments show that a good depth is when the root of the unraveling appears again in the query.
>
> > The proposed approach seems to achieve a lower performance compared to the baseline when evaluated on anchored tree-like queries
> > Questions:
> > Do you have any intuition as to why the proposed approach seems to perform worse, on average, compared to the baseline on anchored tree-like queries?
>
> Our proposed approach does indeed perform worse on anchored tree-like queries if we look at the MRR. Unfortunately we were not able to fully reproduce results from the original GNN-QE paper due to hardware limitations. In our trained GNN-QE the results are closer, while still being slightly better than the /exists GNN-QE.
>
> We believe that this might be due to the way we created unanchored queries: we took the anchored ones and randomly removed anchors, this might have added some unwanted noise in the training and validation dataset and worsen the performance on anchored queries.
>
> We must emphasize: the benefit we obtain for paying this slight price in performance for anchored queries is noticeable: we can now answer unanchored queries with a similar performance, which, in turns, allows for approximating cyclic queries. We feel it is a small price to pay for all we obtain.
>
> > In the experimental setup you mentioned that you "additionally provide a new set of training, validation and test queries...". Is this in addition to the unanchored set originally in the dataset? I was under the impression that your method could only handle unanchored queries?
>
> Our method can handle both anchored or unanchored queries.
> The original datasets don't include unanchored queries, because previous approaches cannot handle them. We build a set of unanchored queries by randomly removing anchors from the original query sets.

---

> > ### Comment · Reviewer_Ajos · 2023-11-23
> >
> > Thank you for you response.

---

### Official Review · Reviewer_x4Ne · 2023-11-04

**Soundness:** 2 fair
**Presentation:** 2 fair
**Contribution:** 1 poor
**Rating:** 3
**Confidence:** 4

**Summary:**

The paper deals with the problem of solving complex queries from knowledge graphs.

**Strengths:**

The idea of approximate a cyclic CQ by a family of tree-like CQs is interesting. The connection with respect to ensemble methods should be discussed. It is not clear how many tree-like queries are used to approximate a cyclic one. This should be stressed in the paper.

**Weaknesses:**

There are many concepts introduced in the paper that are already discussed in inductive logic programming literature. See for instance the definition of containement and homomorphism that are known as substitution in logic programming.

There is a lack of discussion of the related concepts and results known in statistical relational learning and in inductive logic programming. Furthermore, it should be interesting to introduce in the paper the notion of open world assumption that is not discussed.

Please note that the completeness property introduced in the paper corresponds to the notion of clause substitution introduced many years ago in the logic programming literature. The homomorphism introduced in the paper is already called substitution (see fo instance [1]).

Finally, the experimental evaluation should be extended to include other approaches. It is not clear the contribution of the proposed approach.

[1] Stefano Ferilli, Nicola Di Mauro, Teresa Maria Altomare Basile, Floriana Esposito:
A Complete Subsumption Algorithm. AI*IA 2003: 1-13

**Questions:**

Stress the contribution and the experimental results

---

> ### Author Response · Authors · 2023-11-21
>
> Thank you for your comments. We address your concerns below:
>
> >There are many concepts introduced in the paper that are already discussed in inductive logic programming literature. See for instance the definition of containment and homomorphism that are known as substitution in logic programming. There is a lack of discussion of the related concepts and results...
>
> The notions of containment and homomorphism are ubiquitous in computer science and can be considered as basic notions. For example, they are common in database theory, logic and inductive logic. We are just using them here. We note that substitutions somewhat serve a different purpose as it allows us to replace say variables with subexpressions. We are not sure why the reviewer sees the inclusion of these definitions as a weakness. Perhaps the reviewer can comment on this?
>
> > Furthermore, it should be interesting to introduce in the paper the notion of open world assumption that is not discussed.
>
> We are working on incomplete knowledge graphs, which are by definition open world. Perhaps the reviewer can clarify the comment?
>
> > Please note that the completeness property introduced in the paper corresponds to the notion of clause substitution introduced many years ago in the logic programming literature. The homomorphism introduced in the paper is already called substitution (see for instance [1]).
>
> Again, we are not introducing the notions of containment nor homomorphism, but simply using them. These are standard notions in computer science. The connection between containment and homomorphisms is very old and well-known in database theory as we cite in Proposition 3.1.
>
> > Finally, the experimental evaluation should be extended to include other approaches.
>
> We would appreciate it if the reviewer could at least mention one alternative approach for answering cyclic conjunctive queries using state-of-the-art neural symbolic approaches. We are not aware of any other work (see also survey Ren et al. 2023).
>
> > Questions:
> > Stress the contribution and the experimental results
>
> The contributions are clearly presented in the paper (see section “More specifically, our contributions are as follows.” in the Introduction on page 2). More precisely:
>
> - We propose an approximation scheme for complex conjunctive queries using tree-like queries. Moreover, the approximation scheme comes with theoretical guarantees: It is complete in the sense that no false negative query answers are produced. It is optimal in that we provide the best possible approximation using tree-like queries.
>
> - The approximation scheme is adaptive in the sense that it is parameterized by the notion of depth of tree-like queries. For any depth, an approximation exists and higher depth queries potentially provide better approximations. The choice of depth can be tuned depending on available resources, queries and data at hand.
>
> - Our approach is generic and can be used in combination with any neuro-symbolic query processor, provided that unanchored tree-like queries are supported. Figure 1b depicts an unanchored tree-like query in which the input node w is variable. As an independent contribution, we show how to go from anchored to (unanchored) tree-like queries in some neurosymbolic methods.
>
> - We implemented our approach on top of the GNN-QE implementation by Zhu et al. (2022). Results show our techniques are a viable strategy for answering cyclic queries, and that our improvements can be carried over with little cost over this standard neuro-symbolic architecture.
>
> Then in the experiments (Section 5) we describe that our experiments answer the following questions:
>
> (Q1) What is the effect on the performance of answering anchored tree-like queries when the training set includes unanchored tree-like queries as well?
>
>  (Q2) Similarly, what is the effect on the performance of answering general tree-like queries? Looking ahead, as a contribution of independent interest, our results indicate that we can support general tree like queries (Q2) with little or no negative impact for both anchored tree-like queries (Q1). This gives us ground to suggest that general tree-like queries should become default members in training and testing sets of future neuro-symbolic architectures. Our third question relates to our approximation scheme.
>
> (Q3) What is the performance of our approximation scheme in answering cyclic queries? And related, how does this depend on the chosen depth of the unraveling?
>
> These list are taken verbatim from the paper. Could the reviewer be more specific in which contributions and experimental results are not stressed in the paper?

---

### Official Review · Reviewer_FeZT · 2023-11-08

**Soundness:** 2 fair
**Presentation:** 3 good
**Contribution:** 1 poor
**Rating:** 3
**Confidence:** 4

**Summary:**

For a form of conjunctive queries (conjunction of binary predicates, projecting on all-but-one of the variables), this paper applies a technique that works for for the certainty case to reasoning under uncertainty that is inherent in learned models.

**Strengths:**

It is all plausible and I'm willing to accept works for the certainty case. (Except for the infinity claim).

**Weaknesses:**

This paper is trying to apply a technique that works for for the certainty case to reasoning under uncertainty that is inherent in learned models. In particular, it is assuming that the probability of a disjunction is like the maximum probability of its components. Consider the cyclic CQ of Figure 1 (c): as the number of friends of someone goes to infinity, the probability that two of them are coworkers should approach 1. If you wanted a particular x,y and z, what you propose may be more sensible, but not when the query is for just one of them and the others are existentially quantified.

You need to convince us that the sort of queries you can handle is a useful class. (E.g., the valid path restrictions seems very restrictive.) Can it answer *all* queries on knowledge graphs (including when the knowledge graph has arbitrarily many reified relation)?. E.g., this seems to include many fewer queries than could be made with say Problog, which I don't think has any of the restrictions you embrace.

Page 3 "y and z are both existentially quantified" isn't true as it stands. They are universally quantified at the scope of the rule, and existentially quantified in the scope of the body.

I don't understand why "the number of approximations is infinite". If we ground a graph out to propositions (by replacing variables with the elements of the population of entities in all ways), the model is still finite. There is exponential explosion, but it's not infinite. This makes me suspicious. Surely, you can check for loops which would make it finite. However I suspect it is exponential in path length, so that is probably moot. Please give us the complexity.

**Questions:**

What is the mean reciprocal rank of a set? How do you rank sets? If there are multiple witnesses for one x (e.g, multiple instance of y and z for a single x), how do you choose which one if the ground truth? What is the ground truth?

What is "the Spearman correlation rank between the total number of answers...."? Spearman rank correlation measures differences between ranks. Why is it appropriate for the total number of answers?

The MRRs for FB15k-237 seem particularly low. The methods don't seem to work. It seems like the modifications that were made to create  FB15k-237 from  FB15k are exactly what your are exploiting. The Spearman rank correlation seems particularly high. Can you explain these results?

---

> ### Author Response · Authors · 2023-11-21
>
> Thank you for your feedback. We would like to answer to some of your comments below, and also ask for some clarification on your observations.
>
> > This paper is trying to apply a technique that works for the certainty case to reasoning under uncertainty... it is assuming that the probability of a disjunction is like the maximum probability of its components.
>
> We are not assuming the probability of a disjunction is the maximum probability of its components. Perhaps the reviewer can point us to the place in the paper causing the confusion?
>
> > Consider the cyclic CQ of Figure 1 (c): as the number of friends of someone goes to infinity, the probability that two of them are coworkers should approach 1---
>
> We are at a loss with the comment on the infinite number of friends. Perhaps the reviewer can clarify the comment?
>
> > You need to convince us that the sort of queries you can handle is a useful class. Can it answer all queries on knowledge graphs...?. E.g., this seems to include many fewer queries than could be made with say Problog...
>
> The class of **conjunctive queries (CQs) is one of the most fundamental query classes** for structured data, and in particular, for knowledge graphs. As we discuss in the Introduction of the paper, previous machine learning methods can only handle a fragment of the class of CQs (tree-like queries). Here we propose a method for arbitrary CQs (not only tree-like), which has been identified as an **important open problem** in the literature (Ren et al. 2023). Note that the only restriction is having one target variable which is a common restriction in the literature due to scalability reasons.
>
> The notion of **valid paths is used to formalize the notion of unravelings**. It has nothing to do with the expressiveness of the class of queries we are considering. We believe that the reviewer may have misunderstood the intention of valid paths.
>
> We also stress that we are following an active line of research on query answering over incomplete knowledge graphs via **machine learning models (the topic of this conference)**. We do not want to apply symbolic methods (such as Problog) as they tackle different use cases: we see purely symbolic methods as a way of obtaining reliable answers, at a greater cost, while neural approaches provide faster approximations.
>
> > Page 3 "y and z are both existentially quantified" isn't true as it stands. They are universally quantified at the scope of the rule, and existentially quantified in the scope of the body.
>
> This is not correct. The query is a conjunctive query. The output variable is x and the variables y and z are existentially quantified by definition of the query.
>
> > I don't understand why "the number of approximations is infinite"...
>
> The number of approximations is infinite simply by definition, as we can choose arbitrary depths for the unravelings. As explained in the first paragraph of Section 4.1, the approximations are **independent** of the underlying knowledge graph.
>
> > Surely, you can check for loops which would make it finite. However I suspect it is exponential in path length, so that is probably moot. Please give us the complexity.
>
> We do not know what the reviewer means here. Perhaps the reviewer can clarify the comment?
>
> > What is the mean reciprocal rank of a set? How do you rank sets?
>
> This is a standard measure applied in previous work. We compare the set of answers with the log-odds reported by the final layer of the architecture for each node, so that higher odds represent a higher “likelihood”. The mean reciprocal rank reported in the paper is the mean of the reciprocal rank for each query type. Hence, a higher mean reciprocal rank indicates that most probable answers are reported first by the algorithm.
>
> > If there are multiple witnesses for one x (e.g, multiple instances of y and z for a single x), how do you choose which one is the ground > truth? What is the ground truth?
>
> The ground truth consists of the set of all nodes that are witnesses for x in any answer of the query.
>
> > What is "the Spearman correlation rank between the total number of answers...."? Spearman rank correlation measures differences between ranks. Why is it appropriate for the total number of answers?
>
> The measure is between the model cardinality prediction (keeping entities with score over 0.5) and the number of ground truth answers as in GNN-QE or other previous work (betaE, conE).
>
> > The MRRs for FB15k-237 seem particularly low. ... the Spearman rank correlation seems particularly high. Can you explain these results?
>
> Indeed, this also happens in previous work such as [q2b, betaE] and also in the original GNN-QE. The MRR is suggesting that the first right answer appears, on average, in positions 10-11 for unanchored queries, and it varies between 2 and 11 for anchored ones. The spearman’s rank correlation indicates that queries with more ground truth answers, also get a larger number of predicted answers, compared to other queries.

---

> > ### Comment · Reviewer_FeZT · 2023-11-21
> >
> > consider your example q(x) <- Friend(x,y) & Friend(y,z) & coworker(x,z)
> > which is equivalent to
> > \forall x  q(x) <- (exists y \exists z Friend(x,y) & Friend(y,z) & coworker(x,z))
> > which is logically equivalent to
> > \forall x \forall y \forall z (q(x) <- Friend(x,y) & Friend(y,z) & coworker(x,z))
> >
> > You seem to be finding the best y and z for each x. You keep talking about "which approximation to use". (Which is why I said :the probability of a disjunction is like the maximum probability of its components")  I would expect that the person x with the most likely value of q(x) is a person with lots of friends and coworkers; these will satisfy the body just by chance. It's unlikely someone with lots of coworkers and friends will not have any coworkers that are not friends of friends.
> >
> > "We compare the set of answers with the log-odds reported by the final layer of the architecture for each node, so that higher odds represent a higher “likelihood”." - that is the standard way to find the rank of a single prediction. But you are finding the reciprocal rank of a set. My question is simple: how are sets ranked? If I know that, I know how to compute MRR. (Why do we want the rank of a set?)
> >
> > My question about Problog was asking about the expressiveness of queries you can answer.
> >
> > Is the number of paths exponential in the path length? (I'm guessing the answer is yes and that the branching factor is large). Your figures should show this, and there should be some reporting of this.
> >
> > You need to define all the terminology you use. If there is no room to define it, omit it, A paper in ICLR has to be readable by more than the other researchers in a narrow area (whatever the area happens to be).

---

> > > ### Author Response · Authors · 2023-11-21
> > > **Response to reviewer**
> > >
> > > Thank you for your feedback.
> > >
> > > >consider your example q(x) <- Friend(x,y) & Friend(y,z) & coworker(x,z) which is equivalent to \forall x q(x) <- (exists y \exists z Friend(x,y) & Friend(y,z) & coworker(x,z)) which is logically equivalent to \forall x \forall y \forall z (q(x) <- Friend(x,y) & Friend(y,z) & coworker(x,z)) You seem to be finding the best y and z for each x. You keep talking about "which approximation to use". (Which is why I said :the probability of a disjunction is like the maximum probability of its components") I would expect that the person x with the most likely value of q(x) is a person with lots of friends and coworkers; these will satisfy the body just by chance. It's unlikely someone with lots of coworkers and friends will not have any coworkers that are not friends of friends.
> > >
> > > We use the rule-based manner of writing conjunctive queries and the evaluation of conjunctive queries on (knowledge) graphs. The  expression q(x) <- Friend(x,y) & Friend(y,z) & coworker(x,z) logically specifies to find all x in the graph for which there exist a friend y of x and coworker z of z such that also y and z are friends. It is not a matter of finding the best y and z. As a consequence, we do not see the validity of the logical equivalences given by the reviewer.
> > >
> > > > My question is simple: how are sets ranked? If I know that, I know how to compute MRR. (Why do we want the rank of a set?)
> > >
> > > We are not measuring the rank of a set. Instead we are measuring the **rank of the first correct answer**. This is the standard definition of the reciprocal rank where only the rank of the **first relevant answer** is considered, possible further relevant answers are ignored.
> > >
> > > > My question about Problog was asking about the expressiveness of queries you can answer.
> > >
> > > Using our technique we can approximate any unary conjunctive query. Note that the unary part comes from the need to work with vector representation (which is currently the limit of neural methods). We cannot capture full first-order logic with our approximation scheme. Again, sota can only deal with anchored tree-like conjunctive queries.
> > >
> > > > Is the number of paths exponential in the path length?
> > >
> > > Indeed, unravelings are defined in terms of valid paths and there are indeed exponentially many of such paths. It is important that the paths are in the query graph and not the data graph. As such, the branching factor is small since the query graph is very small relative to the size of data. Our experiments show that small unravelngs already give excellent results. We will clarify all this in the revision.
> > >
> > > > You need to define all the terminology you use. If there is no room to define it, omit it, A paper in ICLR has to be readable by more than the other researchers in a narrow area (whatever the area happens to be).
> > >
> > > To our knowledge we define all ingredients needed but will expand on some of them in the revision.  Of course we want our paper to be accessible to a wide audience and do our utmost best to revise it in that way. We stress that our presentation is **like most of the papers in the area of neuro-symbolic query processing** which is an important considerable (and not particularly narrow) area in the machine learning community.

---

> > > > ### Comment · Reviewer_FeZT · 2023-11-21
> > > >
> > > > I see the problem (one of them). You assume that you can treat a learned knowledge graph like a standard knowledge graph. I don't think you can; you need to see it as making probabilistic predictions, which need to be combined using the logic of probability, not the logic of Booleans. If the system is not sure about the truth of some relations/properties, it should not act like it does know the truth.
> > > >
> > > > The paper says "we report the mean reciprocal rank (mrr) of the predicted answer set". That contradicts what you said in your answer (I know what the MRR is, just not the MRR of a set is).
> > > >
> > > > There are lots more things not defined such as (1p/2p/3p/2i/3i/2in/3in/inp/pni/pin)
> > > >
> > > > "Note that the unary part comes from the need to work with vector representation (which is currently the limit of neural methods)". Working with vectors does not imply unary. There is lots of work predicting higher-order relations (some going by the title of "knowledge hypergraphs").

---

### Meta-Review · Area_Chair_ArqQ · 2023-12-09

**Metareview:**

The paper deals with answering probabilistic conjunctive queries on knowledge graphs. Specifically, authors focus on the hard class of cyclic query and propose to approximate them by a family of tree-like queries, and then use results in the database (DB) community to deal with the latter. The authors then propose an extension of an existing GNN (∃GNN-QE) to incorporate such an approximation and test them on classical benchmarks for complex query answering. The proposed ∃GNN-QE is marginally better or better than its baseline on test unanchored queries.

The major concern from reviewers is the difficulty in disentangling the novelty of the contributions and claims w.r.t. known results in the DB literature. Other concerns were raised regarding experimental details and the significance of the results. In the rebuttal, authors were not able convince reviewers concerning the value of their contribution and the great majority (4/5) still voted for rejection (3). I agree with the reviewers that the writing the paper can greatly improve, and make it impactful. I believe the approximation scheme is meaningful and can be useful for KG complex query answering if explained and evaluated with rigor.

Therefore the paper is rejected but I encourage authors to engage with criticism and incorporate it in a new revision of the manuscript.

**Justification For Why Not Higher Score:**

Three reviewers questioned the core contributions and highlighted how the current presentation makes it hard to disentangle what is already known in the DB literature and what is novel in the ML one.

**Justification For Why Not Lower Score:**

N/A

---

### Decision · Program_Chairs · 2024-01-16

Reject